# Healthcare providers' knowledge, attitudes, and perceptions from using targeted sequencing to diagnose and manage drug-resistant tuberculosis (DR-TB) in Eswatini

Maia Madison[1,2,3], Debrah Vambe[1,3], Sein Sein Thi[4], Mangaliso Ziyane[5], Nosisa Shiba[1,3], Babongile Blisset Nkala[1,3], Tara Ness[3,6], Agostinho Viana Lima[7], Sindisiwe Dlamini[5], Siphiwe Ngwenya[4], Anna Mandalakas[3,8], Alexander Kay[1,3]*

1 Baylor College of Medicine Children's Foundation Eswatini, Mbabane, Eswatini, 2 Fulbright U.S. Student Program, New York, New York, United States of America, 3 The Global Tuberculosis Program, Department of Pediatrics, Baylor College of Medicine, Houston, Texas, United States of America, 4 National Tuberculosis Control Program, Manzini, Eswatini, 5 Eswatini, Health Laboratory Services, Mbabane, Eswatini, 6 Department of Biological Sciences, University of Alaska Anchorage, Anchorage, Alaska, United States of America, 7 Centro de Investigação em Saúde de Manhiça, Manhiça, Mozambique, 8 Clinical Infectious Disease Group, German Center for Infectious Research (DZIF) Clinical TB Unit, Research Center Borstel, Borstel, Germany

* Alexander.kay@bcm.edu

## Abstract

Challenges in diagnosing drug-resistant tuberculosis (DR-TB) contribute to a diagnostic gap. Design-locked Targeted Sequencing (TS) assays have the potential to improve DR-TB diagnosis and management. TS assays are now being introduced into low-income, high TB burden settings. Eswatini is among the first high burden countries to have implemented TS for patient care. To evaluate the impact of the current program and optimize future implementation, we evaluated healthcare provider knowledge, attitudes, and perceptions (KAP) of TS and its implementation. We conducted semi-structured interviews with healthcare providers. Interviews were continued until data saturation was reached and analyzed by directed thematic analysis. The study was conducted at 85% of all DR-TB treatment centers (12/14) in rural and urban settings across all four regions in Eswatini. We interviewed nine doctors and eight nurses who were purposively sampled from DR-TB care sites in Eswatini. We found that providers' experience, roles, and settings informed their knowledge and perceptions of DR-TB diagnosis and management. While all healthcare providers wanted to improve comprehensive drug susceptibility testing, operational challenges with the existing program shaped their KAP of TS. In some instances, providers reported that results from TS on sputum improved their ability to provide quality DR-TB patient care. However, they perceived a need for improvements in the delivery of TS results and desired more training to inform their current use of results from sputum and potential future use of results from stool. Overall, healthcare providers

**Data availability statement:** All relevant data are within the paper and its Supporting Information files.

**Funding:** This work was supported by the Fulbright US Student Program and a Dartmouth College John Sloan Dickey Center for International Understanding award titled "Lombard Public Service Fellowship." AK is supported by the Fogarty International Center at NIH (1 K01 TW011482-01). The funders played no role in the conception, design, or completion of the study.

**Competing interests:** The authors have declared that no competing interests exist.

recognized TS as an important new tool with the potential to improve DR-TB patient care. However, they also recognized the need for additional healthcare worker training, community engagement, forecasting to avoid reagent shortages, and enhanced medical information systems. Investments in these areas would likely support more effective and sustainable implementation in Eswatini and other LMICs with high TB burdens.

## Introduction

In 2022, it was estimated that only 43% of incident DR-TB cases were treated globally [1]. Challenges in diagnosing drug-resistant tuberculosis (DR-TB), including rifampicin-resistant (RR-TB), multi-drug-resistant (MDR-TB), and extensively drug-resistant (XDR-TB) contribute to a significant diagnostic gap. TB has traditionally been diagnosed using sputum specimens, but in children and people living with HIV (PLHIV), more invasive procedures are often required to induce sputum production or collect via nasopharyngeal or gastric aspiration [2,3]. Even if specimens are obtained, *Mycobacterium tuberculosis (M. tuberculosis)* can be difficult to detect in specimens from PLHIV and children who often have paucibacillary disease [4–6]. Furthermore, phenotypic drug susceptibility testing (pDST), the current standard of care for diagnosing drug resistance, can delay effective treatment, and is often not available for new DR-TB medications, resulting in empiric treatment and poor clinical outcomes [7,8].

Next-generation sequencing (NGS) has the potential to revolutionize DR-TB diagnosis and management by comprehensively and rapidly detecting mutations in the *M. tuberculosis* genome that are associated with drug resistance [9–11]. While whole genome sequencing of the *M. tuberculosis* genome (WGS) is a promising reference standard for TB diagnosis [12], healthcare, laboratory, and public health professionals are concerned about the implementation of this technology due to cost and complexity [13–16]. The implementation of WGS has been evaluated in a variety of high-income countries (HICs), including the US [16], UK [16], and Canada [14]; and low and middle-income countries (LMICs), including Botswana [13], South Africa [17], and Madagascar [14]. Based on these studies, the acceptability of WGS for DR-TB is generally high among clinicians, public health professionals, and policymakers in HICs and LMICs, with WGS being valued for its ability to provide rapid, comprehensive, and accurate drug resistance profiles. However, stakeholders agreed that there are specific considerations for implementing this technology in LMICs, including the high upfront and maintenance costs, limited accessibility, lack of infrastructure for data sharing, high level of training required, and expensive material and human resources [13,14,16,17].

TB targeted sequencing (TS), which sequences regions of the *M. tuberculosis* genome as opposed to the entire genome, may be faster and more economical than WGS depending on the specimen type used [18,19]. The WHO has endorsed using TS on sputum to identify TB drug resistance [20], but it has yet to recommend

a stool-based method for comprehensive DST [21]. Nonetheless, the WHO has recommended using stool as a first-line diagnostic specimen for Xpert Ultra in children. These guidelines have encouraged the implementation of TS on sputum, and stool TS research to diagnose DR-TB among children and PLHIV in TB high-burden settings [22].

TS has been implemented in a variety of settings, yet most samples that are collected in LMICs, especially in Sub-Saharan Africa, are currently shipped to North America or Europe for sequencing [23]. Consequently, Eswatini's program is leading the way in implementing TS for diagnostic sequencing in a LMIC with high TB burden. In addition to Eswatini, there are in-country TB-TS programs in Benin, Rwanda [24] and Namibia [18].

Due to the small number of programs performing TS in LMICs, there has only been one study describing the implementation of TB-TS [18,23]. In implementing TS in Namibia, they found that TS is faster and cheaper than WGS, but it presents similar logistical challenges, including personnel training, reagent acquisition, sample transport, and limited digital infrastructure. Nonetheless, implementation was made feasible by applying a structured programmatic model for phased implementation and partnerships with international and local organizations [18].

In 2019, with support from the German Ministry of Health, Research Center in Borstel, and Baylor College of Medicine Children's Foundation Eswatini, Eswatini's National TB Control Program (NTCP) developed a protocol for implementing TS at all 14 decentralized DR-TB sites in the country. In this pilot program, TS was introduced into the diagnostic algorithm for TB. Clinical implementation was delayed until September 2021 due to COVID-19 disruptions, procurement challenges, lab personnel training, and structural adjustments of the lab. In 2024, the implementation of TS is firmly established at all DR-TB sites, but currently limited to sputum specimens and sputum culture. Strategic national plans include future stool-based TS to increase the proportion of patients with access to DR-TB testing. Although implementation of TS began in 2021 [25], it is unclear how healthcare providers perceive this technology, and how sequencing on stool will be perceived by clinicians. Crafting a comprehensive strategy to implement TS on sputum and stool requires a context-dependent understanding of the needs and concerns of healthcare providers. We assessed clinicians' knowledge, attitudes, and perceptions (KAP) related to the current use of TS on sputum and its potential use on stool in Eswatini.

## Methods

### Ethics statement

Interviews were conducted according to the principles expressed in the Declaration of Helsinki with written informed consent obtained from all participants. Approval was obtained from all necessary ethical bodies including the Baylor College of Medicine Children's Foundation Eswatini (IORG0006978) and the Eswatini National Human Health Research Review Board, Baylor College of Medicine Institutional Review Board (FWA-00000286), Houston, Texas, USA. All participants gave written or verbal consent.

### Study design

This was a qualitative study investigating clinicians' KAP of TS on sputum and stool for DR-TB diagnosis in Eswatini. Data was collected using semi-structured interviews (SSIs) and analyzed using directed thematic content analysis. This study followed the COREQ checklist for reporting qualitative research (S1 Checklist).

### Context

Eswatini is a low-resource setting with a GDP per capita of US$3,987 [26]. Nonetheless, the country's spending on health is relatively high compared to other LMICs. In 2020, 6.5% of GDP was spent on healthcare, which is above the average spent by LMICs (5.6%), but almost half the amount spent on average by HICs (14%) [27]. Regardless, the country continues to have sub-optimal health outcomes, in part due to persistent poverty and inequality. In 2024, 52.1% of the population lived below the LMIC poverty line [28] and the most recent Gini index in 2016 was 54.6 [26].

In 2022, Eswatini had a high incidence of TB estimated at 325 (95% CI 215–502) cases per 100,000; and a moderate incidence of DR-TB at 19 (95% CI 9.2-28) per 100,000 [1]. Additionally, HIV prevalence in adults was among the highest in the world at 27.4% (27,28). Consequently, Eswatini had a high incidence of TB-HIV coinfection estimated to be 187 (95% CI 104–294) cases per 100,000 [1]. Despite the high estimated TB incidence, only 61% of this total were diagnosed with TB. Limited access to healthcare may contribute to this diagnostic gap because 75% of the population lives in rural areas [29]. To improve access to TB screening in rural areas, the Eswatini Ministry of Health has mobilized "TB Community Champions" to screen for TB.

If an individual is presumed to have TB by a healthcare worker, then specimens are collected and sent via the National Specimen Transport System for testing at one of 35 decentralized laboratory sites throughout the country. TB and RR-TB are primarily diagnosed by Xpert MTB/RIF Ultra (Ultra). If *M. tuberculosis* DNA is detected, then specimens are sent to the National TB Reference Laboratory (NTRL) where *M. tuberculosis* culture, pDST, and line probe assay (LPA) are performed. Although Ultra detects *M. tuberculosis* DNA and mutations in the rpoB gene associated with rifampicin resistance, the majority of rpoB mutations in Eswatini are not detected by Ultra, Mycobacterial growth indicator tube (MGIT) culture or line probe assay (LPA) [30–32].

To address this DR-TB diagnostic gap, in 2019 Eswatini's NTCP integrated TS into the diagnostic algorithm for DR-TB. This pilot program began prior to the updated WHO recommendations about using TS to diagnose DR-TB. To support the implementation of sequencing, the Global Health Protection Program through the German Ministry of Health funded the purchase of two iSeq 100 Illumina Next Generation Sequencing machines, reagents, and the training of laboratory personnel at the National Tuberculosis Reference Lab (NTRL). With support from the Research Center in Borstel and Baylor Foundation, laboratory personnel were trained at the NTRL in 2021 and in 2022. Although the COVID-19 pandemic delayed the procurement of many supplies and the launch of the program, TS was functional at the NTRL in 2021. The Eswatini Ministry of Health implemented the use of TS at all 14 DR-TB sites in Eswatini using a centralized model for laboratory testing. In March 2022, the NTCP created a Clinical Advisory Committee (CAC) comprised of expert clinicians, laboratory personnel and public health officials, to guide clinicians' use of TS results. As part of the CAC mandate, clinicians were offered training on the use of TS results. The implementation of TS was overseen by the technical advisor for the national Programmatic Management of Drug-Resistant Tuberculosis (PMDT) committee.

In this context of DR-TB diagnosis, Eswatini's NTCP delivers DR-TB care in 14 decentralized treatment sites located across 4 regions of the country (Shiselweni, Manzini, Hhohho, Lubombo). Each region has a designated DR-TB doctor to support the clinicians at the treatment sites. In the Manzini region, there are four DR-TB sites; in Hhohho, there are four sites; in Shiselweni, there are three sites; and in Lubombo, there are three sites (Fig 1).

## Sampling and recruitment

Nationally representative purposive sampling was used to select DR-TB providers interviewed for this study. Our intent was to interview clinicians practicing in each region in Eswatini, inclusive of rural and urban environments, and primary and secondary health centers. Physicians and nurses working with DR-TB patients at any of the 14 DR-TB sites were eligible to participate. At a minimum, we aimed to interview one physician and one nurse per region in Eswatini (eight healthcare providers total); at a maximum, we aimed to interview one physician and one nurse at each DR-TB treatment site in Eswatini (21 healthcare providers; seven sites did not have a permanent doctor or nurse). We excluded physicians who were on leave at the time of the study.

Prior to implementation, we discussed and developed the study with the NTCP. We contacted regional DR-TB coordinators to set up interviews with healthcare providers. Regional coordinators, along with experienced clinician-scientists (DV and BB), connected the interviewer (MM) to the DRTB coordinator at the NTCP (a former DRTB care provider), and a total of 20 clinicians (9 physicians and 11 nurses) from all four regions. MM contacted all 21 potential interlocutors. The relations between the researchers and the study population are outlined in S2 Checklist. In obtaining consent, the study details and the researchers' roles were described to the potential participants.

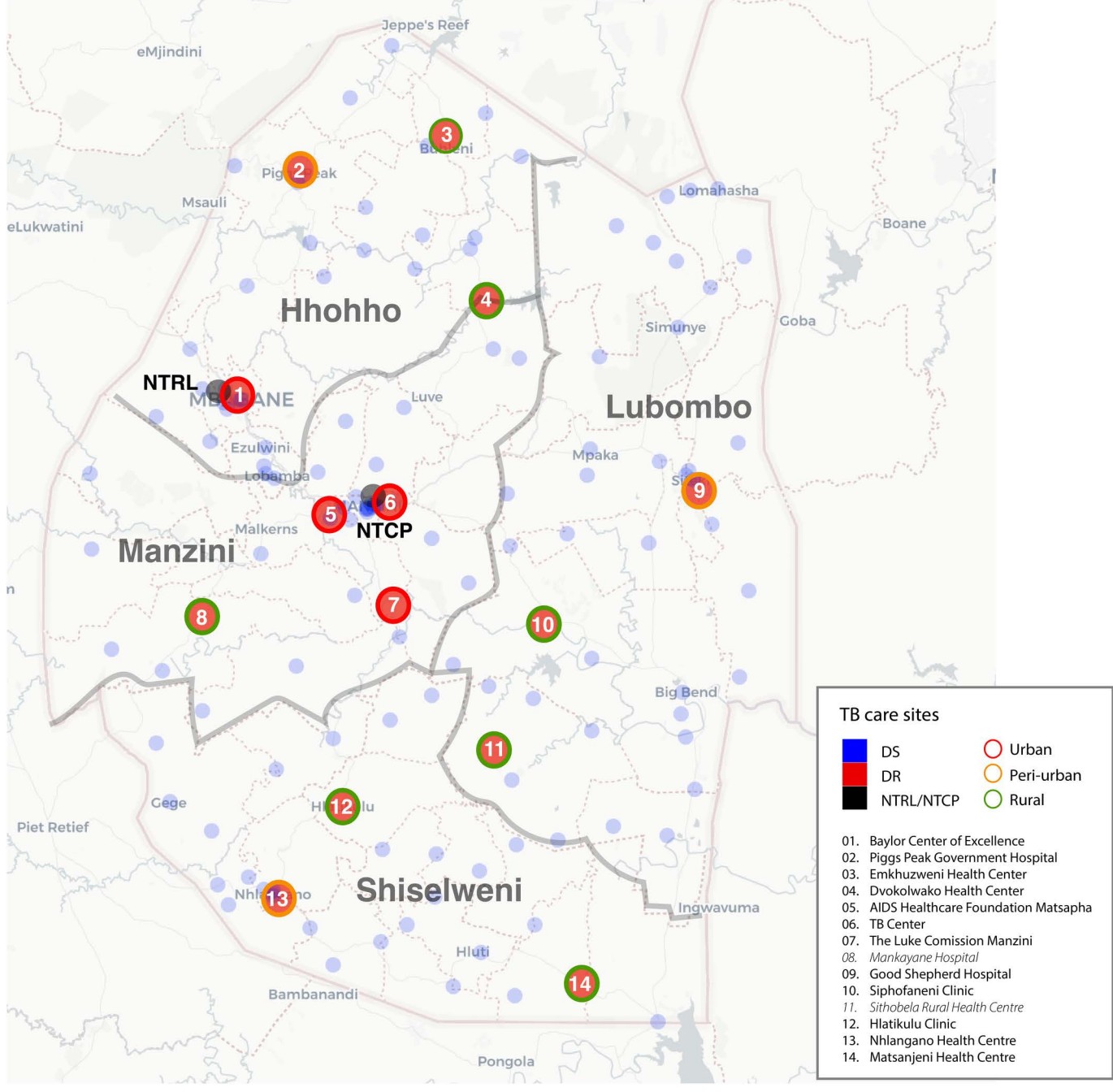

**Fig 1. DR-TB care sites in Eswatini.** Blue circles represent DS-TB sites, while large red circles represent DR-TB sites. Black circles identify the National TB Reference Lab (NTRL) and the National TB Control Program (NTCP). Outer circles indicate the geography classifications urban, peri-urban, and rural. DR-TB sites are labeled in the legend. Interviews were performed at all sites except those in italics in the legend.

## Data collection

We conducted 15 semi-structured interviews with the NTCP coordinator, eight doctors and eight nurses (in two interviews, both a doctor and a nurse were present). Recruitment occurred between January 29, 2024, and March 22, 2024. Written

informed consent was obtained from all participants before participating in and recording the interviews. All interviews were conducted in English for 30 minutes by one researcher (MM) using an SSI guide (S1 Text). Interviews were conducted over encrypted video calls or in a private and quiet location at the participant's place of work. No repeat interviews were conducted. SSIs stopped once data saturation was reached. Notes were taken during the interview and interviews were transcribed verbatim. Transcripts were not returned to participants. After the transcription of the interviews, all audio files were destroyed, and de-identified transcripts were stored on a secure Baylor College of Medicine server. Names were removed from transcripts prior to analysis.

### Analysis

The analysis followed a thematic analysis approach, which is a systematic classification process defined by coding and identifying themes or patterns [33]. The software Taguette 1.4.1-50-geed050b [34] was used to organize and code the transcripts. The codebook contained several initial codes based on the interview topic guides, including KAP of DR-TB diagnosis, TS, and TS on stool. During the preliminary coding process, emergent themes were identified, added to the codebook, and included in the analysis. A second independent coder (AK) coded the transcripts. Codes were compared and any discrepancies were resolved. After the codebook was finalized, the transcripts were re-coded line by line. Coded transcripts were exported from Taguette into Excel. The coded transcripts were further coded for sub-themes and analyzed by MM using Excel (S1 Data). Maps were generated using the R package leaflet (version 1.9.4), and the qualitative schematic and network diagram were created in Adobe Illustrator.

### Results

#### Sociodemographic

Out of 21 potential interlocutors, 18 responded and three nurses did not respond. All clinicians that could be reached consented to being interviewed. One doctor consented to the interview but did not participate due to scheduling around clinical work. In the Hhohho region, 100% (3/3) of doctors and 100% (4/4) of nurses agreed to be interviewed. In the Manzini region, 50% (1/2) of doctors and 50% (1/2) of nurses agreed to be interviewed. In the Shiselweni region, 100% (3/3) of doctors and 100% (2/2) of nurses agreed to be interviewed. In the Lubombo region, 100% (1/1) of doctors and 50% (1/2) of nurses agreed to be interviewed. Ultimately, 16 healthcare workers from 12 DR-TB care sites, and the NTCP coordinator were interviewed (Fig 1). The Hhohho and Shiselweni regions are overrepresented in the study because there are more DR-TB sites in these regions.

   Participants were classified based on the geography of their primary clinic, their specialty training or experience, their involvement in research, and their experience directing or developing healthcare programs in the country (Table 1). The clinical experience among the providers interviewed ranged from less than 10 months to over 13 years. There were 10 female healthcare providers and 7 male providers. All the nurses interviewed were self-identified as TB nurses. All the doctors were caring for TB patients, but some had additional specialized training (Table 1). A greater proportion of doctors had research experience compared to nurses (Table 1). We interviewed an equal number of doctors from rural, urban, and peri-urban areas, while a greater proportion of the nurses interviewed were from rural clinics. Three doctors had experience managing DR-TB programs in the country (Table 1).

#### Themes

During the interviews, several themes emerged regarding the implementation of TS on sputum. These themes include the importance of a well-functioning lab, the need for decentralized services, varying levels of knowledge about TS, challenges facing the result alert system, frustration with workload and delayed results, variable attitudes about when to use TS, the benefits of the Clinical Advisory Committee, and concerns about sustainability. Further, the themes applying to stool specimens included a moderate amount of knowledge of stool as a specimen for TB diagnosis, mixed perceptions of

PLOS Global Public Health

**Table 1. Participant characteristics.**

| | | Doctor, N (%) | Nurse, N (%) |
|---|---|---|---|
| **Specialty** | | | |
| | DR-TB specialist | 3 (33.3) | 0 (0) |
| | TB specialist | 4 (44.4) | 8 (100) |
| | HIV voluntary counseling and testing | 1 (11.1) | 0 (0) |
| | General practitioner | 1 (11.1) | 0 (0) |
| **Geography** | | | |
| | Urban | 3 (33.3) | 2 (25) |
| | Peri-urban | 3 (33.3) | 2 (25) |
| | Rural | 3 (33.3) | 4 (50) |
| **Involved in research** | | | |
| | Yes | 4 (44.4) | 1 (12.5) |
| | No | 5 (55.6) | 7 (87.5) |
| **Programmatic experience** | | | |
| | Yes | 3 (33.3) | 0 (0) |
| | No | 6 (66.7) | 8 (100) |

stool as a specimen, and questioning attitudes towards the feasibility of stool as a specimen. Major themes and selected illustrative quotes are organized in Table 2.

Synthesizing these themes and the sociodemographic information, we found that healthcare providers' clinical, research, and programmatic experiences influence their knowledge of DR-TB diagnosis; healthcare providers perceive challenges in diagnosing DR-TB; operational challenges influence healthcare providers' knowledge, attitudes, and perceptions of TS; and providers are open to implementing TS on stool to diagnose DR-TB if more information is disseminated to them and their communities. These findings are discussed in further detail below.

### Influence of clinical, research, and programmatic experience on healthcare providers' knowledge of DR-TB diagnosis

Healthcare providers' clinical, research, and programmatic experience influenced their knowledge of DR-TB diagnosis. All providers were familiar with RR/MDR/XDR-TB and were highly knowledgeable about their clinical roles in treating and diagnosing these forms of DR-TB. However, providers with more experience caring for DR-TB patients accordingly had greater knowledge of the specific tests and workflow involved in DR-TB diagnosis. One doctor with extensive experience described the diagnostic pathway as follows:

> "All presumptive TB patients who come to our clinic, they give two sputums, one for GeneXpert and one for culture and LPA. So, if you do a GeneXpert, we are using a GeneXpert result saying they are resistant as a proxy to MDR. And sometimes, if you are lucky enough, you get LPA results, which is rare, but maybe after a month, you get culture results which can also confirm INH resistance... We also put them in our genotypic registration. Then, we also have those ones who we diagnose presumptive. Maybe they are close contacts to our current DRTB patients" (P11).

This provider, like several others, had over 5 years of experience treating patients with DR-TB and was involved in quality improvement research. While this doctor described the algorithm in detail, another with nurse with less training and fewer years of experience described the algorithm as follows: "we know from the plan that [when someone] tests positive for TB... we have to run to see if it's a DR, if it's resistant to one drug, then we move on to test for other drugs, but it's clearly

**Table 2. Quotes illustrating themes.**

| Specimen | Theme | Participant quote |
|---|---|---|
| **A) Sputum** | Importance of a well-functioning lab | We used to say that, especially for DRTB, the lab is, in fact, the head and shoulder of DRTB care. (P1/Manzini/Urban/DRTB/Programmatic) |
| | | I don't know if they will improve the turnaround time for the results, because sometimes you find that you have a client, but you can't really start the treatment, because you don't have the results, but you can see the client, and maybe he's not responding well to the treatment (P12/Lubombo/Nurse/Rural/TB). |
| | Decentralized services | I think the first thing was [that] DRTB [was] more centralized. So, issues with access, issues with stigma, because we had a setting hospital which was like it's for DRTB patients. So, we need[ed] to decentralize the services for access but at the same time, we understood there was stigma. So, [we wondered], if [we] decentralize it, are the healthcare workers going to accept that? And the community, how are they going to look at it that there are now DRTB patients around us? [The NTCP] wanted to integrate it with drug-susceptible TB, but it felt like people wanted them to be on their own (P2/Hhohho/Urban/DRTB/Research/Programmatic). |
| | Varying levels of knowledge | Well, so I can just give you the clinical part. That's the only thing that I can explain about the sequence thing (P9/Manzini/Doctor/Urban/TB/Research) |
| | | At first, it was like a push from the [NTCP] program… But now we felt like [healthcare workers] were now like forthcoming... So, we could see them somehow even changing the regimen before the results. They can now confidently talk about sequencing. (P2/Hhohho/Urban/DRTB/Research/Programmatic) |
| | Result alert system | Currently, well, previously, they were sent through emails, and probably if you don't check your emails here and then, you may probably not know whether you've received them or what... So, the patient is almost, it's actually been done treatment, it's only when I'm receiving sequencing results. (P10/Manzini/Nurse/Urban/TB) |
| | Frustration with workload and delayed results | Oof. Sequencing, I think for now, it has increased my workload to some extent, because I find myself changing regimens for patients later on during the treatment. (P11/Lubombo/Doctor/Peri-urban/TB/Research) |
| | | It's just frustrating on the patient side, because now they've been changing treatment about three times now. (P14/Shiselweni/Doctor/Peri-urban/DRTB) |
| | Variable attitudes about when to use TS | If there could be a rollout of the gene sequence to cover maybe all cases from diagnosis, so that at the beginning of treatment, you just have a result that is pointing to the possible mutations that are available and you tailor-make the regimens to what the gene sequence is telling you. It will make life easier. It will also save resources for the country. Because you won't need to start a patient in one regimen and later on change them to another regimen. (P11/Lubombo/Doctor/Peri-urban/TB/Research) |
| | | [We ordered] sequencing for patients whose results in Xpert were MTB detected with resistance detected. Those patients were directly sent for sequencing just for testing to see if they harbor the mutation. Also, those for LPA, first-line LPA, have INH resistance. (P10/Manzini/Nurse/Urban/TB) |
| | | I don't just…wake up in the morning [and] ask for sequencing. No, no, no, no, no. Because the way I understand, it's an exam you cannot just ask like that...We cannot just run sequencing like this. We must make sure that we are really stuck [the patient is not improving]. There is something going on. (P15/Shiselweni/Doctor/Rural/GP) |
| | Benefits of the Clinical Advisory Committee | Even though we might have, you know, those personal feelings that [a] patient may be responding to the previous regimen, we do practice what the committee suggests, because we do think the panel is well experienced and well informed about everything. We consider genome sequencing results. (P14/Shiselweni/Doctor/Peri-urban/DRTB) |
| | Concerns about sustainability | With the sequencing, I'm just hoping it's here to stay. It's not like it will be here for a few years, then it will disappear. Like most of other things that have been introduced before, we find out now they are saying, hey, now the funder is gone. So I'm just hoping it's here to stay because it really, really is helping a lot. (P7/Hhohho/Nurse/Rural/TB) |
| **B) Stool** | Knowledge of stool for TB diagnosis | [I have heard about] diagnosis of TB [from stool], yes, but stool sequencing, no. Okay. I haven't heard of stool sequencing. (P15/Shiselweni/Doctor/Rural/GP) |
| | | It's going to actually help us to improve the pediatric TB case findings so that not only we get limited on DR, first line of DR, but then maybe also having more information on second line drugs as well. Those are the drugs that are not assessed during or when we do stool diagnosis. (P3/Hhohho/Doctor/Peri-urban/TB/Research) |

*(Continued)*

**Table 2.** (Continued)

| Specimen | Theme | Participant quote |
|---|---|---|
| | Mixed perceptions of stool as a specimen | With this stool, [diagnosis] will be quicker… But, the problem, it will be from the patient side. Will they be able to accept the diagnosis based on stool examination? Uh, you diagnose, for example, pulmonary TB using stool, which will be a little bit difficult. And in the area where we are, the problem of belief, with Africans is a little bit, uh, tricky. (P13/Shiselweni/Doctor/Rural/TB) |
| | | I think even healthcare worker perspective is because they don't need to perform this intrusive procedure. They would be more inclined, you know, to prescribe this test. (P1/Manzini/Urban/DRTB/Programmatic) |
| | | There may be reluctance from the healthcare workers, because to be honest, very few people would be comfortable collecting stool every day. It's something that even me, I wouldn't want to. So, there are cases, there are stool samples that might not be collected, because maybe the healthcare worker on that day was not in the mood to enter stool. (P11/Lubombo/Doctor/Peri-urban/TB/Research) |
| | Feasibility of stool as a specimen | And maybe the capacity of the lab, because at this stage, we are struggling doing GeneXpert on sputum. So, I'm looking to say, is the lab going to be able to handle these samples? I don't know, and I don't think so. So, yeah, I think there might be quite a few issues. (P11/Lubombo/Doctor/Peri-urban/TB/Research) |

outlined" (P12). The different levels of detail in these descriptions highlight discrepancies in providers' levels of knowledge of the specific tests involved in DR-TB diagnosis.

Additionally, providers described their patients as predominantly adult males living with HIV. Few had experience treating children with TB. As a result, most providers were more knowledgeable about the challenges facing adults and PLHIV with DR-TB than children. Further, clinicians involved in research and directing programs had a greater knowledge of trends in DR-TB diagnosis and TS in the country than clinicians who only focused on providing care. Most clinicians without research experience were aware of ongoing research on DRTB and TS in the country but didn't have knowledge of specific research activities.

## Healthcare providers perceive challenges in diagnosing DR-TB

Many providers recognized GeneXpert as a vital tool for TB diagnosis, but a limited one for DST. For example, one doctor with research experience noted: "Sometimes GeneXpert misses the Rifampicin-resistance pattern according to the mutation, I think I491F, something like that? – the lab can explain to you very well about that" (P9). While not all clinicians mentioned the Isoleucine 491 Phenylalanine (I491F) mutation, which is associated with rifampicin resistance and is not detected by GeneXpert, LPA, or MGIT, most clinicians highlighted that GeneXpert was limited by the fact that it detected "RIF-resistance only" (P1, P2, P5, P7, P8, P11, P13).

Beyond the limitations of diagnostic tools, healthcare providers underscored the pivotal role of a well-functioning laboratory. One provider noted, that "we used to say that, especially for DRTB, the lab is, in fact, the head and shoulder of DRTB care" (P1). For example, if the lab doesn't send results to clinicians in a timely fashion, then patient care suffers. All providers emphasized that even if GeneXpert, culture, or LPA results are accurate, they take too long, likely due to human and material resource shortages, leaving the clinician to treat the patient empirically. A nurse noted that this was a common occurrence: "I don't know if they will improve the turnaround time for the results, because sometimes you find that you have a client, but you can't really start the treatment, because you don't have the results, but you can see the client, and maybe he's not responding well to the treatment" (P12). Healthcare providers exhibited varied attitudes toward updating empiric treatment based on delayed GeneXpert, culture, and LPA results. Some said they adopt a precautionary approach by treating the result that presents the "worst-case scenario," while others said they rely more on clinical judgment to manage patients: "we will consider the clinical [factors], as I told you, we consider the patient exposure, the kind of exposure,

the time of exposure, the clinical examination, chest X ray, and then we associate with other tests. Then I take my decision" (P9).

Additionally, clinicians suggested that geography presented additional barriers to DR-TB diagnosis. Some clinicians highlighted that the prior physical separation of DR-TB and DS-TB services may have exacerbated stigma and discouraged individuals from seeking timely treatment. One doctor noted that the initial centralization of DRTB care led to "issues with stigma, because we had a hospital setting which was like [only] for DRTB patients. So, we need[ed] to decentralize the services for access… We wanted to integrate it with drug-susceptible TB, but it felt like people [healthcare workers and the community] wanted them to be on their own" (P1). Furthermore, providers in rural areas emphasized that a lack of resources and the transport of specimens from rural clinics to a centralized laboratory can lead to delays in result reporting that impact their ability to provide the most effective treatment.

Despite this challenge, clinicians serving rural areas stressed how decentralizing the DR-TB program has made services more accessible. An unintended consequence of this decentralization has been to increase the independence of physicians at distant sites. Consequently, one physician with programmatic experience perceived clinicians in Eswatini as highly competent and commended their ability to manage DR-TB cases. Nonetheless, most healthcare providers perceived obstacles to DR-TB management that their clinical skills could not overcome, necessitating technical and systems-level change.

### Influence of operational challenges on healthcare providers' knowledge, attitudes, and perceptions of TS

Since sputum-based TS was introduced in Eswatini, operational challenges have shaped providers' knowledge, attitudes, and perceptions of each step in the implementation pathway (Fig 2). First, providers have gained varying levels of knowledge, mostly related to its clinical applications. For example, one doctor (P9) stated that "I can just give you the clinical part. That's the only thing that I can explain about the sequence thing" and another nurse described "what I understand is that with sequencing, it's a further step which is more detailed compared to the GeneXpert, the line probe assay, and the culture and DST. It gives more details about the resistance." While nearly all the healthcare workers described their knowledge of TS in terms of its utility as a diagnostic tool, one nurse described his knowledge of why TS is a more effective diagnostic: "what I know is that there's whole genome sequencing where we use the entire genome for the TB… the DNA of the microbe, where we're looking at resistant patterns throughout the entire genome in order to identify any mutations rather than being restricted to certain regions where in this country [there are] certain mutations" (P10). This nurse was referring to the I491F mutation, which is prevalent in Eswatini and not detected by traditional pDST or any tests completed on the GeneXpert platform. Similarly, other clinicians mentioned that TS is more effective than Xpert Ultra or phenotypic liquid DST in identifying this resistance mutation, without naming the specific amino acids.

Although we didn't examine healthcare providers' knowledge of TS before its introduction in Eswatini, clinicians state that they have gained knowledge through NTCP workshops and experience with using TS clinically. Seven of the interviewed healthcare workers mentioned attending a "workshop" to learn about TS for DR-TB diagnosis. One doctor described that they learned about sequencing at the workshop, but that "you can't integrate all those things in one day or two days" and the learning is "about what you call actually progressive training, progressive learning"(P15). That progressive learning seems to have come through practical clinical experience. The program director at the NTCP noted that they had seen an improvement in knowledge and the use of results from clinicians over the course of the last two years: "now we feel like [physicians] are forthcoming. [They will come to us and say]—"we have a client who's not doing well—could it be a mutation that we are not picking up?" …[Physicians] can now confidently talk about sequencing" (P1). Providers' exposure to and use of TS has increased their knowledge of TS which has increased their comfort in using the technology (Fig 3a).

Although providers' experiences have increased their knowledge, their perceptions of TS have been mixed due to operational challenges (Fig 2). Most providers described result delivery as inconsistent because of machine failures,

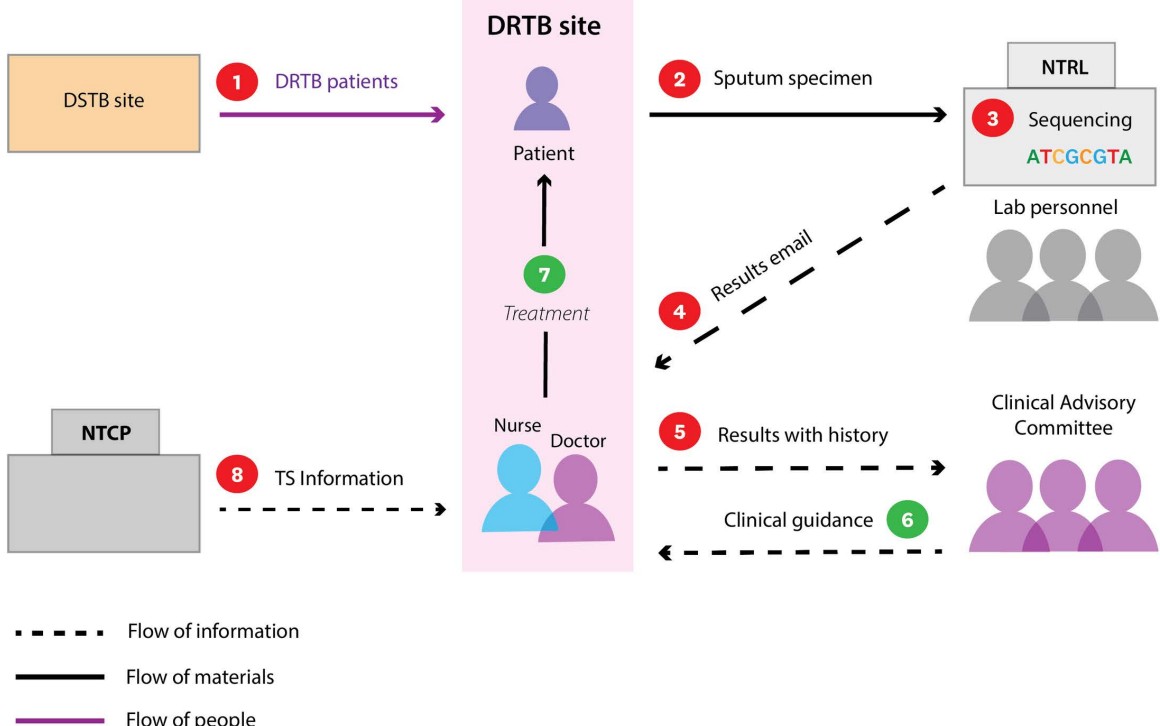

**Fig 2. Implementation of TS.** The schematic outlines perceived challenges and successes of sputum-based TS implementation. Numbers indicate areas where healthcare workers perceived strengths (green) and weaknesses (red) in the implementation pathway. 1) Case detection is limited in communities due to stigma and centralized DRTB care sites. 2) Delayed arrival of sputum specimens at the National TB Reference Lab due to difficult specimen collection and problems with transportation. 3) Delayed and inconsistent results generation due to limited reagents and laboratory personnel. 4) Delayed action on results due to a lack of a laboratory alert system. 5) Increased workload and frustration due to paperwork and multiple regimen changes. 6) Improved patient care because of clinical guidance from the clinical advisory committee. 7) Improved patient care because of more accurate resistance predictions and improved turnaround times compared to traditional modalities. 8) Need for more information to be disseminated by the National TB Control Program.

human error, shortages of resources, and unreliable result delivery. For example, providers mentioned that effective treatment decisions are often delayed because results are sent by email and there is no standardized result alert system. A nurse said "if you don't check your emails here and there, you may not know whether you've received [the results]... So, the patient is almost, it's actually done [with] treatment, it's only then that I'm receiving sequencing results" (P10). In this system, it's easy to miss results that could inform more effective treatment. One clinician said that recently the laboratory addressed this problem by calling physicians in advance of sending the results. Despite this isolated report of improvement, most clinicians attributed patient frustration with treatment regimen changes, and clinician frustration with an increase in their workload to delayed TS result reporting. For example, one doctor remarked: "it's just frustrating on the patient side, because now they've been changing treatment about three times now."

Due to the operational challenges limiting the use of TS, some clinicians have limited experience with TS. Clinicians' amount of experience with the technology and their perception of resource availability appear to inform their comfort with the tool and confidence in the results. While providers had received some information about TS, most did not consider themselves to have "specialized training." Consequently, clinicians expressed different attitudes about when to use TS to inform patient care, from using it early and often to as a last resort. One doctor believed that TS should be used for all patients— "you['d] just have a result that is pointing to the possible mutations that are available and you tailor-make

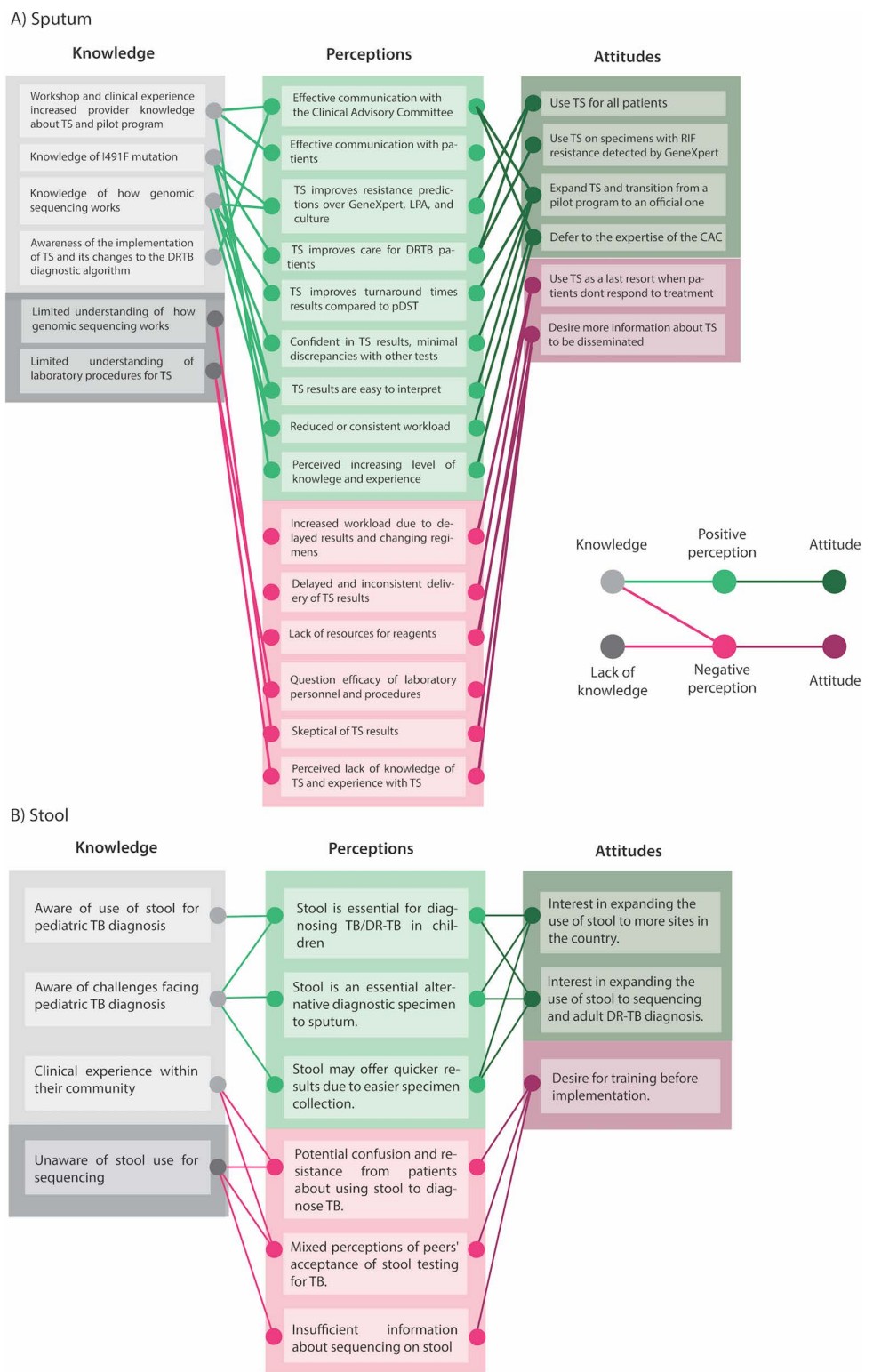

**Fig 3. Network diagram of clinicians' knowledge, attitudes, and perceptions of using TS for DRTB diagnosis.** A) sputum-based TS. B) stool-based TS. Light grey nodes represent knowledge held by clinicians, while dark grey nodes represent a lack of knowledge. Light green nodes and edges represent positive perceptions, while light pink nodes and edges represent negative perceptions. Dark green and maroon edges and nodes represent attitudes shaped by positive or negative perceptions.

the regimens" (P11). At the other end of the spectrum, two clinicians thought that TS should only be used as a last resort when patients don't respond well to treatment— "we must make sure that we are really stuck" (P15), or "make it a last resort, but use it" (P12). Between these attitudes, the majority of clinicians believed that TS should be used as a second test after rifampicin resistance is detected by GeneXpert (P6, 9, 10, 14) (Fig 3A).

Further, when TS identified additional resistance compared to GeneXpert, LPA, or MGIT, clinicians with limited TS experience preferred to rely on clinical judgment, while those with more experience used TS results. Despite differences in opinion about when to use TS, most clinicians found TS results easy to interpret and appreciated the expertise of the clinical advisory committee when they were unsure about results. One doctor noted: "even though we might have, you know, those personal feelings that the patient may be responding to the previous regimen… we do practice what the committee suggests, because we do think the panel is well experienced and well informed about everything. We consider genome sequencing results." Providers unanimously highlighted the clinical advisory committee as improving DR-TB care (Fig 2).

Although obstacles have challenged the implementation of TS, most providers commented that TS improves DR-TB diagnosis. Clinicians noted improvements in the turnaround time of results compared to pDST and improved resistance predictions. They emphasized that these improvements in DR-TB care are worth the added work. Accordingly, providers were open to the expansion of TS from a pilot to an official program, but they wanted to ensure its sustainability, by evaluating its cost-effectiveness, improving the timeliness of results, and training more clinicians and laboratory personnel (Fig 2).

**Providers are open to implementing TS on stool to diagnose DR-TB if more information is disseminated to them and their communities**

Although most providers did not have experience diagnosing TB in children, they were aware of the use of stool as a specimen for diagnosing TB in children (Table 2B; Fig 3B). Notably, providers perceived stool as an essential specimen for TB diagnosis in children, providing an alternative where traditional specimen types may fail, and offering quicker results due to easier collection. While providers knew less about TS on stool, they perceived it as a promising tool to increase drug resistance detection using this specimen type (Table 2B).

Providers anticipated potential confusion and resistance among patients regarding stool-based diagnosis due to traditional beliefs about stool testing (Table 2B). Similarly, there were mixed perceptions among providers about their peers' acceptance of stool-based testing for DR-TB. Some believed that clinicians would appreciate the introduction of this specimen to reduce invasive testing, while others questioned healthcare providers' willingness to collect stool (Table 2B).

In addition to disagreement about the acceptability of using stool, providers felt there was insufficient information about the feasibility of TS on stool. While clinicians deferred the assessment of technical feasibility to laboratory personnel, they were concerned about the operational feasibility of implementing such a test given the challenges they identified with the current TS implementation (Table 2B).

Despite these concerns, most providers were open to the use of stool for TS. Providers noted the potential benefits of stool-based testing for pediatric DR-TB diagnosis and expressed interest in expanding its use to diagnose adults. Nonetheless, they mentioned a desire for more information and comprehensive training before implementation (Fig 3B).

## Discussion

We assessed clinicians' KAPs of using TS on sputum and stool for the diagnosis of DR-TB. We found that clinicians perceive TS on sputum as a valuable tool for improving DR-TB diagnosis and they are open to implementing TS on stool. While providers recognize the potential benefits of sequencing on stool, they are concerned about patient and healthcare provider acceptance and a lack of information regarding effectiveness. For TS on sputum, providers are also concerned about limited resources, delayed results, the lack of an alert system, and inadequate information. Addressing these concerns will be crucial for implementing TS for DR-TB diagnosis in pediatric and adult populations in TB high burden settings where health systems are often constrained.

We found that providers had an in-depth clinical understanding of using TS to diagnose DR-TB but varied and mostly limited understanding of the technical aspects of the technology. Generally, providers with greater levels of sub-specialization and research or programmatic experience had greater knowledge of TS and were more open to using it. This phenomenon does not appear to be context specific. In 2018, clinicians in Canada and Madagascar were found to have limited experience with integrating TB NGS into clinical care and were generally more comfortable with using previously established diagnostic tools like GeneXpert [14]. Further, in South Africa in 2022, providers were concerned that less experienced providers could struggle with the technology if they weren't provided with training [17] . These examples are indicative of the paradigm that knowledge is important for trust, and trust is important for the adoption of new interventions [16]. It seems that with more experience and training, providers gain comfort with TS and are more open to implementing it. The Clinical Advisory Committee seemed to increase provider confidence with implementation in Eswatini and may be a model that could increase knowledge, trust, and adoption of this technology in other settings.

Like other settings, providers in Eswatini were open to implementing TS for TB diagnosis but expressed an interest in obtaining more information. Across varied settings, stakeholders want to receive evidence that sequencing is effective and cost-effective [12–14,16,17]. Clinicians in Madagascar wanted to receive evidence that sequencing had additive diagnostic yield over the current tools like GeneXpert [14]. In Botswana, there was support for WGS if all stakeholders were engaged including patients, healthcare staff, lab workers, and the public. South African clinicians were "ready and willing" to implement sequencing, but they would be more comfortable explaining results to patients if they had additional training [17]. Stakeholders want to be well-informed before adopting and implementing sequencing technologies; this desire for additional training was shared by providers in Eswatini.

DR-TB providers in Eswatini expressed excitement to learn about TS and highlighted the importance of gathering with peers to feel united against the threat of DR-TB. In Eswatini, in-person training hasn't been conducted broadly because of the cost and limited human resources. Although in-person training may reap benefits over time, it requires upfront costs that may be prohibitive for National TB control programs [12]. Globally, stakeholders in government and policy-level positions are concerned that a lack of training due to resource limitations may pose a barrier to implementing WGS [16]. In Eswatini, these concerns seem to apply to TS, but to a lesser extent because of well-established virtual meeting platforms. Online tutorials may be useful interactive training that avoids the costs of in-person training [13]; however, these modalities may garner less engagement because providers prefer in-person instruction [15]. Furthermore, virtual training may be constrained by inadequate internet and computer access. A combination of in-person and virtual training may present a solution that minimizes costs, augments community engagement, and builds trust in sequencing technologies [35].

Providers in Eswatini were also concerned that TS might be too expensive to implement widely and sustainably. This concern made several providers hesitant to use sequencing unless it was necessary. Concerns about cost are not limited to Eswatini. In South Africa and Madagascar, clinicians interviewed about the use of WGS for TB diagnosis held similar concerns about the cost of sequencing [14,17]. It is important to note that WGS is more expensive and more complicated to implement clinically than TS. Even in high-income settings, stakeholders were concerned about the cost-effectiveness and scalability of WGS [12–16].

Moreover, most sequencing in LMICs is funded by governments and international aid organizations such as the WHO, USAID, UKAID, Wellcome Trust, and the European and Developing Countries Clinical Trials Partnership (EDCTP). In Madagascar, WGS is funded by the international Global Fund to fight AIDS, Tuberculosis and Malaria and partnerships with international and domestic academic institutions. In Eswatini, TS is funded externally by the German Ministry of Health. Furthermore, foreign funding is not unique to TS. Many LMIC NTCP activities are supported by international organizations. In Madagascar, public health officials raised concerns about dependence on external funders and policy makers, however none of the clinicians interviewed in Eswatini raised concerns about reliance on foreign governments and non-governmental organizations.

Although funding for TS is concentrated in high income countries, the geographic landscape for diagnostic sequencing appears to be broadening. Before the COVID-19 pandemic, most studies evaluating stakeholders' perceptions of NGS-based TB diagnostics were conducted in high-income, low-TB prevalence settings [14–16]. In the wake of the COVID-19 pandemic, similar studies have been conducted in low-resource high TB prevalence settings [13,17]. Although the COVID-19 pandemic delayed the implementation of TS in Eswatini, it also necessitated massive upscaling of sequencing for viral surveillance globally [36]. The expanded infrastructure for sequencing worldwide may bolster low-resource settings' capacity to implement sequencing for TB diagnosis, however it may also exacerbate disparities in genomic sequencing between high and low-income countries [37]. To address this concern, health personnel suggest that specific attention should be given to increasing capacity for sequencing in LMICs, including expanding digital infrastructure for equitable data storage and sharing [13,14,16].

In Eswatini, providers raised concerns about a lack of digital infrastructure. Many DR-TB sites have the capacity for a laboratory information system, but not all of them, and TS results have not been integrated into this system. Consequently, providers noted that TS results were frequently delayed and lost due to the lack of a standardized reporting and data storage system. Expanding electronic medical records to rural and urban clinics equally and broadly enhancing informatics infrastructure will be necessary for storing and sharing sequencing data.

Some of these implementation challenges can be addressed during a preparatory phase prior to implementation. For example, prior to implementing TS, a team in Namibia had a yearlong preparatory phase during which they identified sites for TS implementation, established a timeline, planned the scope, ethics contracts, budgeting, human resources, laboratory protocols, workshops, and defined the main outcomes of the intervention and strategies to assess outcomes [18]. Eswatini prepared similarly for the implementation of TS; however, many of the challenges identified with TS implementation were challenges that are reflected in the larger laboratory system, particularly around laboratory result reporting. Additionally, this qualitative study was not planned as a measure of outcomes during a preparatory phase. Consequently, a limitation of the study is that providers were not interviewed prior to the implementation of TS, so we cannot compare their KAP before and after implementation. Further, while we interviewed healthcare providers at DR-TB sites, healthcare providers at DS-TB sites and laboratory personnel may have different perspectives. Additionally, we reached saturation before interviewing a physician or nurse at two of the rural sites. Due to the exploratory design of the study, the results provide insight into healthcare providers' KAP of TS and may inform implementation in Eswatini and elsewhere, but they are not conclusive guidelines for implementation.

In conclusion, we assessed clinicians' knowledge, attitudes, and perceptions of using TS on sputum and stool for the diagnosis of DR-TB. We found that providers had variable experiences with and knowledge of using TS to diagnose MDR-TB. Several providers had extensive knowledge and experience, while others had limited experience due to reagent shortages and delayed results, which hampered their knowledge and trust in TS. Consequently, providers in Eswatini were concerned that TS might be too expensive to be implemented sustainably. Additionally, providers were concerned about the inconsistent time to result for TS and the lack of a standard result-sharing and data storage system. Nonetheless, providers were open to using TS on sputum and stool and enthusiastic about its potential impact on patient care. To implement TS effectively in TB high-burden, resource-constrained settings, capacity needs to be bolstered by training healthcare providers, engaging communities, reducing reagent shortages, and improving medical information systems.

## Supporting information

**S1 Checklist. COREQ checklist.**
(DOCX)

**S2 Checklist. Inclusivity in global research questionnaire.**
(DOCX)

**S1 Text.** **Semi-structured interview guide.**
(PDF)

**S1 Data.** **Thematic coding.**
(XLSX)

## Acknowledgments

We thank all healthcare providers who volunteered to be interviewed for the study.

## Author contributions

**Conceptualization:** Maia Madison, Debrah Vambe, Anna Mandalakas.

**Data curation:** Maia Madison.

**Formal analysis:** Maia Madison, Alexander Kay.

**Funding acquisition:** Maia Madison.

**Investigation:** Maia Madison.

**Methodology:** Maia Madison, Debrah Vambe, Agostinho Viana Lima, Anna Mandalakas, Alexander Kay.

**Project administration:** Maia Madison, Debrah Vambe, Sein Sein Thi, Mangaliso Ziyane, Nosisa Shiba, Babongile Blisset Nkala, Sindisiwe Dlamini, Siphiwe Ngwenya.

**Resources:** Nosisa Shiba, Babongile Blisset Nkala.

**Software:** Maia Madison.

**Supervision:** Debrah Vambe, Sein Sein Thi, Mangaliso Ziyane, Nosisa Shiba, Babongile Blisset Nkala, Alexander Kay.

**Validation:** Maia Madison, Alexander Kay.

**Visualization:** Maia Madison.

**Writing – original draft:** Maia Madison, Alexander Kay.

**Writing – review & editing:** Maia Madison, Debrah Vambe, Sein Sein Thi, Mangaliso Ziyane, Nosisa Shiba, Babongile Blisset Nkala, Tara Ness, Agostinho Viana Lima, Sindisiwe Dlamini, Siphiwe Ngwenya, Anna Mandalakas, Alexander Kay.

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
