## [Decision Letter · Decision Letter 0]

PGPH-D-24-02931

Healthcare providers’ knowledge, attitudes, and perceptions from using targeted sequencing to diagnose and manage drug-resistant Tuberculosis (DR-TB) in Eswatini

Dear Dr. Kay,

Thank you for submitting your manuscript to PLOS Global Public Health. After careful consideration, we feel that it has merit but does not fully meet PLOS Global Public Health’s publication criteria as it currently stands. Therefore, we invite you to submit a revised version of the manuscript that addresses the points raised during the review process.

Please note that we have only been able to secure a single reviewer to assess your manuscript. Please see the reviewer's comments below. We are issuing a decision on your manuscript at this point to prevent further delays in the evaluation of your manuscript. Please be aware that the editor who handles your revised manuscript might find it necessary to invite additional reviewers to assess this work once the revised manuscript is submitted. However, we will aim to proceed on the basis of this single review if possible. 

Could you please revise the manuscript to carefully address the concerns raised?

We look forward to receiving your revised manuscript.

Kind regards,

Steve Zimmerman, PhD

PLOS Staff Editor

Journal Requirements:

2. Please provide separate figure files in .tif or .eps format.

3. In the online submission form, you indicated that “Data are available upon request to the corresponding author.”.

a. In a public repository,

b. Within the manuscript itself, or

c. Uploaded as supplementary information.

Additional Editor Comments (if provided):

Reviewers' comments:

Reviewer's Responses to Questions

**Comments to the Author**

1. Does this manuscript meet PLOS Global Public Health’s publication criteria ? Is the manuscript technically sound, and do the data support the conclusions? The manuscript must describe methodologically and ethically rigorous research with conclusions that are appropriately drawn based on the data presented.

Reviewer #1: Yes

2. Has the statistical analysis been performed appropriately and rigorously?

Reviewer #1: N/A

3. Have the authors made all data underlying the findings in their manuscript fully available (please refer to the Data Availability Statement at the start of the manuscript PDF file)?

Reviewer #1: No

4. Is the manuscript presented in an intelligible fashion and written in standard English?

Reviewer #1: Yes

5. Review Comments to the Author

Reviewer #1: This manuscript presents a well-structured qualitative study exploring the knowledge, attitudes, and perceptions of healthcare providers regarding the use of targeted sequencing (TS) for diagnosing DR-TB in Eswatini. The study addresses an important gap in implementation research, as TS is increasingly recognized as a valuable tool for TB diagnosis in Eswatini. The manuscript is well-written, methodologically sound, and provides a clear synthesis of the data collected. However, there are several areas where improvements could be made to enhance clarity, depth, and contextualization.

Introduction

1. You mentioned that in some high income countries and LMICs, WGS has been implemented for diagnosis of DR-TB. Are there specific cases where targeted sequencing (TS) has been implemented for DR-TB diagnosis? If so, can you provide examples?

2. Can you provide a summary of the key challenges and facilitators of implementing targeted sequencing in LMICs specifically? Given that the barriers faced by LMICs and high-income countries often differ, highlighting these LMIC-specific factors would help contextualize the feasibility and scalability of the technology in resource-limited settings. Also, it would be good to highlight the perception or acceptability of WGS in the countries where it has been implemented, if the studies you cited explored these aspects.

3. Was TS implemented in Eswatini in 2022 across the country? Or was it piloted in several sites first? Can you please add this information in the introduction?

Methods

1. It is good that you give background information about the TB burden and diagnosis in Eswatini. Can you add more information about the implementation of TS by the National TB Control Program? For example, was the clinical guideline for DR-TB diagnosis updated and disseminated across the regions? Were doctors and nurses provided with formal training? What preparatory steps were taken to facilitate the implementation of TS in Eswatini?

2. Can you clarify which levels of the healthcare system were covered in this study? How were the TS results integrated into clinical decision-making, and how did they influence the treatment plans for DR-TB patients?

3. Can you provide clear inclusion/exclusion criteria for participant selection (like years of experiences, specialty, etc.)? I noticed that in some regions you recruited more participants, was there any rationale behind it?

4. Are you aware of the Consolidated criteria for reporting qualitative research (COREQ), which is a commonly used checklist for reporting qualitative studies. Did the reporting of results follow the COREQ checklist or any other established guidelines for qualitative research reporting?

Results

1. What about participants’ knowledge of TS specifically? Did they have knowledge about TS prior to the implementation? Was any training provided to improve their knowledge about TS or TB diagnosis in general?

2. The organization of the main results can be improved for clarity. For example, does Table 2 covers all the main themes in the results section or only the “Healthcare providers perceive challenges in diagnosing DR-TB” section? Also in the section "Healthcare providers perceive challenges in diagnosing DR-TB," themes are presented in a table with participant quotes but without further explanation. Since it is likely that multiple participants commented on each theme, summarizing the overall patterns or trends observed across participants would provide a clearer and more informative synthesis of the findings.

Discussion

1. In countries where TS or WGS was implemented, were there specific preparatory steps taken before implementation? If so, what lessons can Eswatini learn from these experiences to enhance the successful adoption and integration of TS into its healthcare system?

2. Since cost was discussed, can you provide more details on who covers the cost of testing in the countries you cited? Are TS or WGS tests funded by governments, international donors, or private institutions? How is the cost of TS managed in Eswatini, and what funding mechanisms are currently in place to support its implementation?

6. PLOS authors have the option to publish the peer review history of their article (what does this mean? ). If published, this will include your full peer review and any attached files.

**Do you want your identity to be public for this peer review?** For information about this choice, including consent withdrawal, please see our Privacy Policy .

Reviewer #1: No

---

## [Editor Report · Decision Letter 1]

Healthcare providers’ knowledge, attitudes, and perceptions from using targeted sequencing to diagnose and manage drug-resistant Tuberculosis (DR-TB) in Eswatini

PGPH-D-24-02931R1

Dear Dr. Kay,

We are pleased to inform you that your manuscript 'Healthcare providers’ knowledge, attitudes, and perceptions from using targeted sequencing to diagnose and manage drug-resistant Tuberculosis (DR-TB) in Eswatini' has been provisionally accepted for publication in PLOS Global Public Health.

Best regards,

Leeberk Raja Inbaraj, MD

Academic Editor